# Modeling the Impact of Unreported Cases of the COVID-19 in the North African Countries

**DOI:** 10.3390/biology9110373

**Published:** 2020-11-03

**Authors:** Salih Djilali, Lahbib Benahmadi, Abdessamad Tridane, Khadija Niri

**Affiliations:** 1Laboratoire d’Analyse Non Lineaire et Mathamatiques Appliquées, Universite de Tlemcen, 13000 Tlemcen, Algeria; s.djilali@univ-chlef.dz; 2Mathematic Department, Faculty of Exact Sciences and Informatics, Hassiba Benbouali University, 02180 Chlef, Algeria; 3Department of Mathematics and Computer Science, University Hassan II Ain Chock, Casablanca 20000, Morocco; 900210781@uaeu.ac.ae (L.B.); khniri@yahoo.fr (K.N.); 4Department of Mathematical Sciences, United Arab Emirates University, Al Ain P.O. Box 15551, UAE

**Keywords:** COVID-19, unreported cases, lockdown, basic reproduction number, hospitalized individuals

## Abstract

**Simple Summary:**

One of the challenges facing the countries to contain the COVID-19 is to trace people that were in contact with an infected person. Failing to identify the possible infected people leads to unreported cases of the COVID-19, which results in massive infection among the population and even superinfection events. In this work, we study the impact of the lockdown implemented by three North African countries on reducing the infections in the pandemic’s first wave. Then, we investigate the effect of the unreported cases in the increase of the number of infected people when each country relaxed the population’s mobility in the “Eid” period, resulting in the second wave of the COVID-19.

**Abstract:**

In this paper, we study a mathematical model investigating the impact of unreported cases of the COVID-19 in three North African countries: Algeria, Egypt, and Morocco. To understand how the population respects the restriction of population mobility implemented in each country, we use Google and Apple’s mobility reports. These mobility reports help to quantify the effect of the population movement restrictions on the evolution of the active infection cases. We also approximate the number of the population infected unreported, the proportion of those that need hospitalization, and estimate the end of the epidemic wave. Moreover, we use our model to estimate the second wave of the COVID-19 Algeria and Morocco and to project the end of the second wave. Finally, we suggest some additional measures that can be considered to reduce the burden of the COVID-19 and would lead to a second wave of the spread of the virus in these countries.

## 1. Introduction and Model Construction

One of the major problems facing public health makers is tracking people who contact a person with COVID-19. Such people might not be traceable all the time if they are unaware of being infected (i.e., asymptomatic), or if they have mild symptoms and choose not to take the COVID-19 test and to be reported. This issue could create a massive infection in the population if the control measures are not respected (such as wearing the face mask and social distancing). In fact, in the course of the COVID-19 pandemic, we witnessed superspreading [1,2] events in several countries due to a single unreported case ( see, for example, in [3,4,5]).

Many countries used different approaches to identify infected people, such as massive testing. However, not every country has the meaning to test the maximum possible of their populations. Therefore, it is difficult to understand how unreported the COVID-19 cases contribute to the dynamic of the spread of this ongoing pandemic and the best public health control measures needed to contain this issue.

To study this problem, we aim to investigate a mathematical model of the unreported cases of the COVID-19 in three north African countries: Algeria, Egypt, and Morocco. We choose these countries because they belong to the same geographical region, North Africa, and their population similarity. However, each country is dealing with COVID-19 testing differently. From the available data in Our World Data Website [6], only Morocco has reported the number of tests performed every day. As far as the testing policies, Egypt and Algeria test people with symptoms and key groups (i.e., key workers, admitted to hospital, came into contact with a known case, and returned from overseas).

For Morocco, the testing policies are open public testing, which also includes the asymptomatic. For the tracing policies, public health in Morocco follows the “Limited” tracing. However, Algeria and Egypt are using the “Comprehensive” tracing, which means all cases are traced. Our study does not include Libya, due to the lack of data about the COVID-19, and Tunisia because the epidemic seems to be under control. Since the beginning of the COVID-19 pandemic, a wealth of mathematical models studied different aspects of the spread of the COVID-19; we mention a few [7,8,9,10,11,12,13,14,15,16,17]. In contrast to the mathematical model of reported and unreported COVID-19 infections by Magal and Webb [8], we predict, in this work, the reported and unreported cases by taking into consideration the mobility of the population of three countries, which is provided in [18,19]. These data allow justifying our choice of the right transmission rate of the disease in each country. For this purpose, we proposed the following system of differential equations
(1)$$\left\{\begin{array}{c} \dot{S}\left(t\right)=-\beta \left(t\right)S\left(t\right)\left(U\left(t\right)+E\left(t\right)\right),\\ \dot{E}\left(t\right)=\beta \left(t\right)S\left(t\right)\left(U\left(t\right)+E\left(t\right)\right)-\mu E\left(t\right),\\ \dot{I}\left(t\right)=p\mu E\left(t\right)-\left(\gamma +\rho \right)I\left(t\right),\\ \dot{U}\left(t\right)=\left(1-p\right)\mu E\left(t\right)-\gamma U\left(t\right),\\ \dot{H}\left(t\right)=\rho I\left(t\right)-mH\left(t\right), \end{array}\right.$$
with the following initial conditions
$$S\left(0\right)=S_{0},\;\;\;\;E\left(0\right)=E_{0},\;\;\;\;I\left(0\right)=I_{0},\;\;\;\;U\left(0\right)=U_{0},\;\;\;\;H\left(0\right)=H_{0},$$
where S(t),E(t),I(t),U(t), and H(t) stand, respectively, for the susceptible population, exposed and asymptomatic individuals, reported infection cases, unreported infection cases, and hospitalized individuals due to the COVID-19 disease at time *t*. The incubation period for the COVID-19 is denoted by 1/μ=5 days; 1/γ=10 days is the average of the infection period for the COVID-19 [7,10]. 1/ρ=5 days represents the average of the period for an infected individual to stay in the infected stage (non declared infection case) for the COVID-19 infection until a complication appears. On the other hand, we assume that 80% of the infection cases are reported (p=0.8) (see in [8]). An infected individual remains in hospital for medical care from 10 to 22 days, depending on the individual’s age and possible comorbidity. Here, we consider that this average to be 1/m=15 days. The transmission rate functional is denoted by β(t) where it is assumed to take the following form.
(2)$$\beta \left(t\right)=\left\{\begin{array}{c} \tilde{\beta },\;\;\mathrm{for}\;\;t\in \left[0,T\right],\\ \tilde{\beta }\exp^{-\varepsilon (t-T)},\;\;\mathrm{for}\;\;t>T. \end{array}\right.$$

The time *T* stands for the date of implementing an efficient lockdown that leads to a decrease in the number of active infection cases. The rate ε represents the degree of efficiency of this restriction, where ε=0 shows that this restriction is inefficient. The transmission functional β(t) in the suggested form Equation (Equation 2) represents the effects of the lockdown on the outbreak of the COVID-19 disease.

The justify of our choice of β(t) is motivated by the mobility data and the data of reported cases in each country as presented in Figure 1. In fact, the decline of mobility in public places and its increase in the residential area shows the level of adherence of the population to mobility restriction implemented by each country. On the other hand, the increasing part corresponding to Morocco and Algeria, can be expressed in our model by a constant transmission rate $\tilde{\beta }$ and the decreasing part can be expressed by the decreasing transmission where the degree of decrease is measured by rate ε, i.e., the efficiency of the governments restriction. The analysis of the data for Egypt is different from the other two countries. In this case, the active infection cases data continue to rise, which correspond to constant transmission rate $\beta \left(t\right)=\tilde{\beta }$ (or ε=0).

To achieve the above goals, we arrange the paper in the following form. In the next section, we approximate the main parameters of our predictions and highlight the countries that have controlled the spread of the COVID-19 disease. Then, we project the peak of the active cases and the end of the epidemic in each of the three countries. In Section 3, we propose some measures that help to reduce the burden of the epidemic, such as increasing the number of tests per day to identify the unreported cases. Furthermore, the influence of these tests on predicting the epidemic peak is investigated. A conclusion section ends this paper.

## 2. Approximation of the Parameters

Based on WHO and Johns Hopkins University reports [20,21], we obtain the time series of the evolution of the reported active infection cases Figure 1. Our goal is to understand how the evolution of the number of individuals asymptomatic stages affects the number of symptomatic persons. We will then give the approximative number of people in these stages using a system Equation (Equation 1). We will also connect the data obtained in Figure 1 and the mobility reports presented by Google [18] and Apple [19].

We start with Egypt. From Figure 1, it is clear that the infection is increasing exponentially. This means that the control measures are not well respected. Although the mobility report in [18] shows a reduction of mobility of −50% in public places, −2% in grocery and pharmacy, −26% in parks, −44% in transit stations, and −13% in workplaces, there is +12% increase in the places of residence. We apply the least square method to estimate the parameter β(t). In fact, we can consider a constant transmission functional $\beta \left(t\right)=\beta_{0}$. For the initial conditions, we assumed that all the population is susceptible, i.e., $S_{0}=9.842\times 10^{7}$. For the reset of the initial, we used following values, $E_{0}=785$, $I_{0}=160$, $U_{0}=128$, and $H_{0}=160$. The best fit for $\tilde{\beta }$ is $\tilde{\beta }=1.98\times 10^{-9}$, ε=0. Moreover, the basic reproduction number corresponding to this country is $R_{0}=1.3641$ (Appendix A). Figure 2 shows that by 14 June there was ∼10,000 unreported COVID-19 cases and ∼48,000 asymptomatic cases. These predictions could an explanation to why the number of cases continue to rise in Egypt after 14 June.

In the case of Morocco, from the mobility report [18], we clearly see the mobility restriction implemented by this country had lead more reduction of population mobility compared to Egypt, with −54% in shops and leisure, −23% in food and pharmacies, −41% in parks, −51% in transit stations, −31% in workplaces, and +20% for the places of residence. These data reflect the impact of the state of medical emergency (lockdown) that Moroccan authorities declared on 20 March. On 9 June, the lockdown was relaxed in some less affected regions with the partial opening of the economy. The data in Figure 1 reflect this situation. We also find that the turning point which starts from T=92 (8 April), where it shows the result of the complete restriction of the social movements, which reduces the active infection cases immediately. Based on this facts we approximate the parameters, where we find p=0.8, γ=1/10, μ=1/5, m=1/15, and ρ=1/5. For the initial condition, we used the total population of Morocco $S_{0}=3.603\times 10^{7}$. For the rest of the initial values, we used the following approximations $E_{0}=694$, $I_{0}=35$, $U_{0}=50$, and $H_{0}=35$. For the transmission rate, we got $\tilde{\beta }=6.4\times 10^{-9}$ and ε=0.0105. As shown in Figure 3, the epidemic peak is reached at t=102, starting from 22 January 2020 with 3447 infection cases. At the peak, the unreported infection cases are approximated by 8690. Furthermore, notice that on 14 June the number of unreported cases is around 1000 and that of asymptomatic cases is ∼2000. The basic reproduction number corresponding to Morocco found to be $R_{0}=1.6141$. The following figure shows the results of our approximations.

Although we could not find the mobility report for Algeria, there are similarities between Algeria and Morocco in the timing of the control measures taken by both countries. In fact, the Algerian authorities started the lockdown on 12 March, but it was partially relaxed on 13 June in a similar fashion to Morocco. Further, from on Figure 1, we can tell that there is partial respect to restriction of the population mobility in Algeria, which can be placed in the same level of controlling the spread of the COVID-19 as Morocco. However, we find the efficacy of this restriction was less compared to Morocco. Our fitting shows T=110, which is around 29 April. The estimation parameters give us p=0.8, γ=1/10, μ=1/5, m=1/15, and ρ=1/5. For the initial conditions, we choose $S_{0}=4.223\times 10^{7}$, which is Algeria’s actual population. The data give the following approximations to the rest of the initial conditions $E_{0}=615$, $I_{0}=38$, $U_{0}=30$. The transmission rate is $\tilde{\beta }=4.35\times 10^{-9}$ with ε=0.0043. The epidemic peak will be reached at t=120 starting from 20 April 2020 with 3211 cumulative infection cases (green color figure). Besides, the unreported infection case will be approximated by 1506 active infection cases. Moreover, the basic reproduction number corresponding to this country is $R_{0}=1.2859$. Figure 4 shows the fitting of data and the prediction of the other variables of our model.

**Remark** **1.**
*We should notice the efficiency of the lockdown rate ε for Morocco is greater than Algeria, which indicates that the Moroccan authorities had more success in controlling the population mobility.*


### Predictive Results

We are now interested in approximating the time of the end of this pandemic in North African countries. For Egypt, and based on the data in Figure 1 and Figure 2, we can deduce that the turning point is not yet there. Therefore, we can predict that the infection cases will continue to rise. By expanding the time in the previous figure, Figure 2, we predict the evolution of the infection cases in Figure 5. Therefore, we can highlight that symptomatic cases (I(t)) will disappear at t=400, this shows that it 260 days of epidemic starting from 14 June 2020 remain. The highest number of symptomatic active infection cases is about $2\times 10^{6}$. The pressure on hospitals is also high where it will reach its highest number at t=276 with $5.53\times 10^{6}$ active infection cases. Moreover, the unreported cases will rich a high number with $9.68\times 10^{5}$ active infection cases. More details are offered in the following figure.

The scenario in Morocco is different, where this country exhibits an effective lockdown. As shown in Figure 6, the highest number of infection cases is reached at t=97 with 3447 active infection cases at t=102 starting on 22 January 2020. Our simulation shows that the individuals in the asymptomatic stage are 4322 individual at t=102. Hospitalized individuals are also approximated by 8683 individuals at the same time. The unreported cases are also reached at this peak time by 1594 infection cases. By our simulation, the epidemic in Morocco would have end by 7 August 2020. More illustrations are highlighted in the following figures.

The evolution of the number of active infection cases in Algeria is similar to Morocco. In fact, Algeria was also on its way to contain the spread of the COVID-19 disease, where there is a turning point in the curve of the number of symptomatic infection cases as it is highlighted in Figure 4. Hence, from Figure 7, we estimate that the epidemic in Algeria should end after 110 days from 14 June 2020 (i.e., 1 October 2020 ). Moreover, the higher number of symptomatic infection cases is reached at t=120 with 3230 infection cases. The unreported cases reach the highest number at t=127 with 1506 infection cases. At t=40, the higher number of hospitalized individuals is reached with 8361 individuals. More illustrations are offered in the following figure.

The restriction of human mobility by the lockdown has a clear effect on the reducing the number of infected individuals. This fact can be seen thorough the difference between the prediction of Algeria and Morocco cases compared to Egypt. The lockdown reduced the epidemic peak and the predicted time of the end of this epidemic in these North African countries. Figure 8 shows that in the absence of the lockdown, the number of reported infected cases will have similar exponential growth as Egypt. However, maintaining the number of reported infected cases after relaxing the lockdown is a big challenge to Algeria and Morocco. The current increase in the number of the COVID-19 cases in these two countries in mainly due to the timing of relaxing the lockdown and management of the COVID-19 after the lockdown aftermath, particularly with respect to the unreported cases.

## 3. The Impact of the COVID-19 Testing Efficacy in Egypt

With the relatively late lockdown in Egypt, which was implemented by the end of May, and the early relaxation at the end of June, the number of deaths due to COVID-19 in Egypt was the highest in Africa [22]. One major issue in containing this epidemic is the low number of COVID-19 tests performed (4000 per day) [22]. This problem would obviously increase the number of unreported cases. In fact, one of the solutions that proposed by Peto [23] to end the epidemic is to have mass testing facilities. In this section, we propose to study the impact of increasing the COVID-19 testing capacity in Egypt and the effect of such a measure on reduce the disease outcome on the public. For this purpose, we will augment our model Equation (Equation 1) to the following system,
(3)$$\left\{\begin{array}{c} \dot{S}\left(t\right)=-\beta S\left(t\right)\left(U\left(t\right)+E\left(t\right)\right),\\ \dot{E}\left(t\right)=\beta S\left(t\right)\left(U\left(t\right)+E\left(t\right)\right)-\left(\mu +T\left(t\right)\right)E\left(t\right),\\ \dot{I}\left(t\right)=\left(p\mu +T\left(t\right)\right)E\left(t\right)+T\left(t\right)U\left(t\right)-\left(\gamma +\rho \right)I\left(t\right),\\ \dot{U}\left(t\right)=\left(1-p\right)\mu E\left(t\right)-\gamma U\left(t\right)-T\left(t\right)U\left(t\right),\\ \dot{H}\left(t\right)=\rho I\left(t\right)-mH\left(t\right), \end{array}\right.$$
where T(t) represents the rate of the revealed individuals by a test of the COVID-19 disease. We presume that this function takes the following special form,
(4)$$T\left(t\right)=\left\{\begin{array}{c} 0,\;\;for\;\;t\in [0,M],\\ T^{*},\;\;for\;\;t>M, \end{array}\right.$$
where $T^{*}$ refers to the additional quantity of tests for revealing infected asymptomatic and unreported individuals, and 1/T is the average time spent by an infected individual in each class before being diagnosed positively by the COVID-19 disease (which are in E-class or U-class). Moreover, *M* represents the time initiating the massive population testing for the COVID-19. The remaining parameters have the same epidemiological meaning of the model Equation (Equation 1).

We can see clearly, from our simulations (Figure 9), that as we increase *T*, the peak of the epidemic delays slightly. However, we found that by increasing the test capacity in Egypt ( from T=0 to T=0.01), we reduce the number of unreported cases (*U*) by 42% and asymptomatic individuals (*E*) by 36%. Moreover, we can also observe a reduction in the number of hospitalized people by 30%. Therefore, by increasing Egypt’s testing capacity to the maximum, it is possible to control the COVID-19.

## 4. The Influence of Relaxation of the Measures on the Spread of COVID 19 in the North African Countries

The relaxation of the North African countries’ control measures was about a month before an Islamic holiday, “Eid al-Adha”, which we will refer to as the Eid. The Eid was celebrated on the 30–31 July. From the last week of June, the North African countries’ population start preparing for the celebration by shopping and traveling to celebrate with their families. The decision to relax the lockdown and the restriction on population mobility, in this particular time, comes with a huge increase in the COVID-19 cases. This increase in the number of reported cases is reflected by the data that can be seen clearly through Figure 10 and Figure 11. In this section, we will treat the cases differently (see in [24] in terms of the quality of the decisions made by the three governments and their influence on the spread of COVID-19 disease:**Algeria**:In the previous section, our simulation showed that the COVID-19 epidemic in Algeria would be finished by the end of September, which has been highlighted clearly in Figure 10 by the figure in blue. However, because of the wrong time of relaxing the population mobility, the actual data highlighted in red points in the same figure suggest a different outcome. Indeed, as shown in Figure 10, there is an increase in infection cases (stage 3). After this holiday, the rate decreased slightly (stage 4). The relaxation of measures was inevitable because the lockdown made a huge economic deficit. It left no choice for the decision-makers to reduce the severity of the measures and try to live with the fact that COVID-19 will continue spreading in the Algerian community. The corresponding transmission rate that considered in the fourth stages in Figure 10 is expressed in the following manner,
$$\beta \left(t\right)=\left\{\begin{array}{c} \tilde{\beta }\;\;for\;\;t\leq T_{1},\;\;\mathrm{Stage}\;1,\\ \tilde{\beta }\exp^{-\varepsilon (t-T_{1})}\;\;for\;\;T_{1}<t\leq T_{2},\;\;\mathrm{Stage}\;2,\\ {\tilde{\beta }}_{1}\;\;for\;\;T_{2}<t\leq T_{3},\;\;\mathrm{Stage}\;3,\\ {\tilde{\beta }}_{2}\;\;for\;\;T_{3}<t,\;\;\mathrm{Stage}\;4, \end{array}\right.$$
where $T_{1}=110\;\;T_{2}=148,\;\;T_{3}=191$, and ${\tilde{\beta }}_{1}=4.25\times 10^{-9}$, ${\tilde{\beta }}_{2}=3.66\times 10^{-9}$. In Figure 12, we can deduce that this decision will have a huge impact on the evolution of the infection cases, which needs to be revised again before losing control of the spread of this contagious disease. We predict a higher decrease in the first weeks of October because reopening the schools and universities on 20 September 2020 will lead to more acute infection cases.**Morocco**:As the data show, the similarity between the evolution of the infection cases in the two countries (Algeria and Morocco) can be seen clearly in Figure 10 and Figure 11, where it is obtained that the COVID-19 disease spreads in this country through four principal stages as it is highlighted in Figure 11. Morocco’s situation is similar to Algeria, where the Moroccan authorities decided to relax the population mobility before the Eid, which led to an increase in the number of acute infection cases. Based on these mentioned points, we can write the transmission rate as
$$\beta \left(t\right)=\left\{\begin{array}{c} \tilde{\beta }\;\;for\;\;t\leq T_{1},\;\;\mathrm{Stage}\;1,\\ \tilde{\beta }\exp^{-\varepsilon (t-T_{1})}\;\;for\;\;T_{1}<t\leq T_{2},\;\;\mathrm{Stage}\;2,\\ {\tilde{\beta }}_{1}\;\;for\;\;T_{2}<t\leq T_{3},\;\;\mathrm{Stage}\;3,\\ {\tilde{\beta }}_{2}\;\;for\;\;T_{3}<t,\;\;\mathrm{Stage}\;4, \end{array}\right.$$
where $T_{1}=90\;\;T_{2}=140,\;\;T_{3}=202$, and ${\tilde{\beta }}_{1}=5.35\times 10^{-9}$, ${\tilde{\beta }}_{2}=4.35\times 10^{-9}$. It can be seen clearly that there is a continued increase in infection cases, as highlighted in Figure 13. If the disease’s progress goes in the same way, we also expect that the infection will disappear by the end of 2021.**Egypt**:The case in Egypt is entirely different than Algeria and Morocco. The relaxation of the measures in the Eid period did not make any difference in the number of cases. Our only explanation is the lack of reported cases. The evolution of infection cases follows three stages, as it is highlighted in Figure 14. It worth mentioning that there is a decrease in the number of in the period of Eid; this decrease continued until mid-September. If the situation continues in the same way, the infection will disappear by the end of 2020 or the beginning of 2021 as it is shown in Figure 15. Using Figure 14, we can consider that the transmission rate takes the following formulation,
$$\beta \left(t\right)=\left\{\begin{array}{c} \tilde{\beta }\;\;for\;\;t\leq T_{1},\;\;\mathrm{Stage}\;1,\\ {\tilde{\beta }}_{1}\;\;for\;\;T_{1}<t\leq T_{2},\;\;\mathrm{Stage}\;2,\\ {\tilde{\beta }}_{1}\exp^{-\varepsilon (t-T_{2})}\;\;for\;\;T_{2}<t,\;\;\mathrm{Stage}\;3, \end{array}\right.$$
where ε=0.0105,$T_{1}=132$, $T_{2}=169$, $\beta_{1}=5.35\times 10^{-9}$.

Table 1 shows the basic reproduction number of each country in each stage of the disease.

For Morocco, we obtained that the variance is equal to ≈311, equal to ≈379 in the case of Egypt, and ≈170 in the case of Algeria.

## 5. Discussion and Concluding Remarks

The coronavirus epidemic outbreak happened in the North African countries of Egypt, Morocco, and Algeria at the beginning of March. The reported infection cases in these countries were just a part of a larger number of real infection cases, including infected individuals with no symptoms. Here, we considered a model that investigated the approximation of this category of the infected class. Further, the number of individuals under medical care was considered. We used the available data [20,21] from the beginning of the world pandemic until 14th, 2020to estimate fit the curve of infected (reported) cases and to estimate the parameters of the model. Consequently, we predicted the number of unreported and hospitalized people in each country. By looking at the mobility data (from Morocco and Egypt) and the similarity between these countries (specifically Algeria and Morocco) in the measure taking, we analyzed the efficacy of the lockdown measure. Applying lockdowns was not easy for the authorities; respecting these restrictions, by populations, vary from one country to another. The evolution of the disease in these North African countries is different, which was reflected in the large difference in predictions, as so we draw the following conclusions.

**Egypt**:The restrictions in this country were not well respected. This conclusion is based on the evolution of the infection cases presented in Figure 1 where all the infection cases (asymptomatic, reported, unreported, and hospitalized) continued to rise. This led us to claim that if not measure is taking, the infection will not disappear any time soon (see Figure 5). As the lockdown and restriction of human mobility can be costly on Egypt’s economy, it is important to find alternative measures to contain the spread of the COVID-19. For this purpose, we suggested having massive testing for the population. Augmenting the number of tests per day is one of the most useful tools to reduce the size of the epidemic and lower the burden on the public health facilities, Figure 9 shows that with *T* = 0.01, can reduce the number of unreported cases by 42% and hospitalize by 30%. As we could not track the number of tests performed by the Egyptian authorities, by increasing *T* to a level the fit the data after 14 June to 9 August, we could get the level of the infected case as of today. We also investigate the impact of the Eid holiday on the progress of the disease in Egypt. Our study showed no effect of the relaxation of the restriction on human mobility on the number of cases.**Morocco**:Up to relaxing the lockdown, Morocco was doing great in obligating citizens to respect the lockdown. The infection cases were going remarkably down, and the population mobility in the public places was substantially reduced. We also predicted that the infection would disappear from the Moroccan community by 7 August 2020. The lockdown in Morocco was more effective than one done in Algeria and Egypt, which is reflected by the high rate of the lockdown efficiently ε=0.0105. It was proved that the lockdown had a huge role in reducing the number of infection cases in this country and reducing hospitals’ pressure. Without the restrictions, there would have been a drastic increase in infection cases, as highlighted in Figure 8 as the restriction had a very bad influence on the economy. Therefore, many countries, including Morocco, tried to ease its lockdown by removing some bans on the population mobility, which led to an unavoidable increase in infection cases due to the mobility’s augmentation. In fact, the authorities relaxed the restriction one month before the Eid, and the country could not manage the aftermath of the lockdown (See Figure 11), and the country is having a second wave of the COVID-19 disease. As of the 14 June, the number of unreported cases was over 1000 and the number of asymptomatic was around 3000. By increasing the number of tests COVID-19 capacity in the country, it is unavoidable to see a huge increase in the number of the COVID-19 cases. With this trend of daily increase of cases, our simulation projects the end of 2021 ( see Figure 13).**Algeria**:As mentioned before, this country is similar to Morocco. It exhibited an effective lockdown with less efficiency (ε=0.0043). We predicted that, if the aftermath of the lockdown was managed adequately, the COVID-19 epidemic would end by 1 October 2020. In a similar fashion, the Algerian authorities tried to relax the lockdown by reducing the severity of the restriction population mobility between regions and opening stores and parks in the cities that did not report any infection cases. However, such a decision should be strictly observed by the population and mandate everyone to follow the government guideline. The relaxation of the restriction before the Eid was the reason for the second wave of the COVID-19 (See Figure 10). In fact, by 14 June, the number of unreported cases was over 1500, and the number of asymptomatic was around 3300). By increasing the number of tests COVID-19 in Algeria, we see a continued increase in the number of the COVID-19 cases. Our simulation projects the end of 2021 ( see Figure 12).

In summary, although the restriction of population mobility at the beginning of the COVID-19 pandemic harmed the economies of the North African countries studied herein, it has been beneficial in reducing the increase of infection cases. Removing the lockdown should be planned to maintain low endemicity and protect the population from unreported COVID-19. This can be achieved by having strict guidelines to manage the aftermath of a lockdown and increase the testing during the lockdown and after the lockdown. Taking such strict protection measures while enforcing social distancing and face masks in public places would help to avoid a second wave of COVID-19, which would exhaust the public health capacity.

## Figures and Tables

**Figure 1 biology-09-00373-f001:**
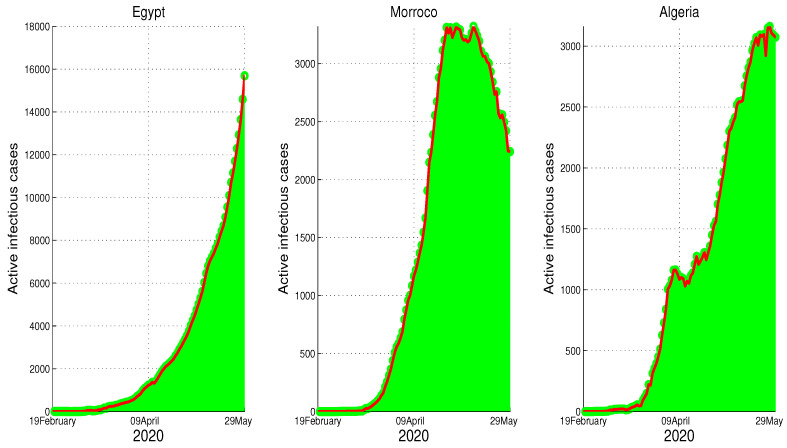
The active infection cases in Egypt, Morocco, and Algeria between t=0 (22 January 2020) and t=144 (14 June 2020).

**Figure 2 biology-09-00373-f002:**
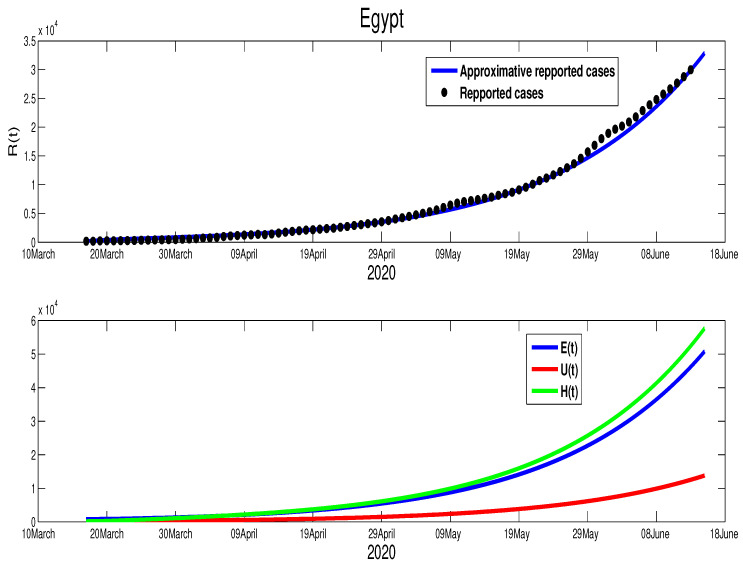
The fitting of data of active COVID-19 cases in Egypt and corresponding times series of the other variables.

**Figure 3 biology-09-00373-f003:**
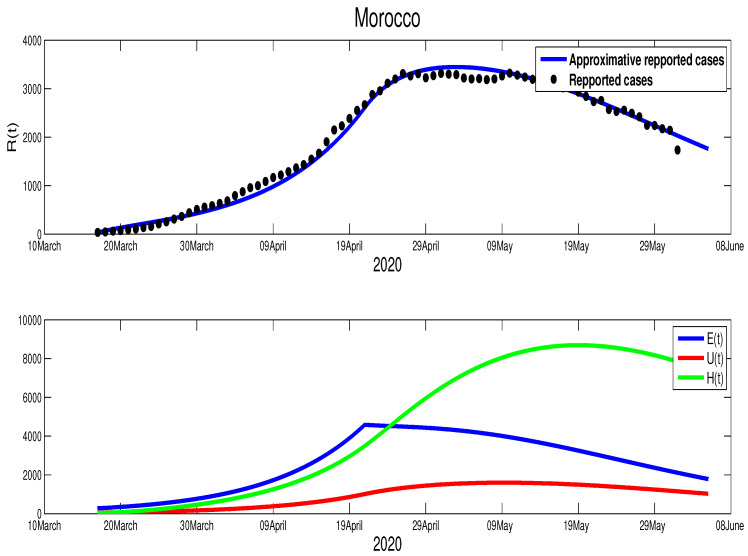
The data fitting of the infected reported the spread of the COVID-19 disease in Morocco.

**Figure 4 biology-09-00373-f004:**
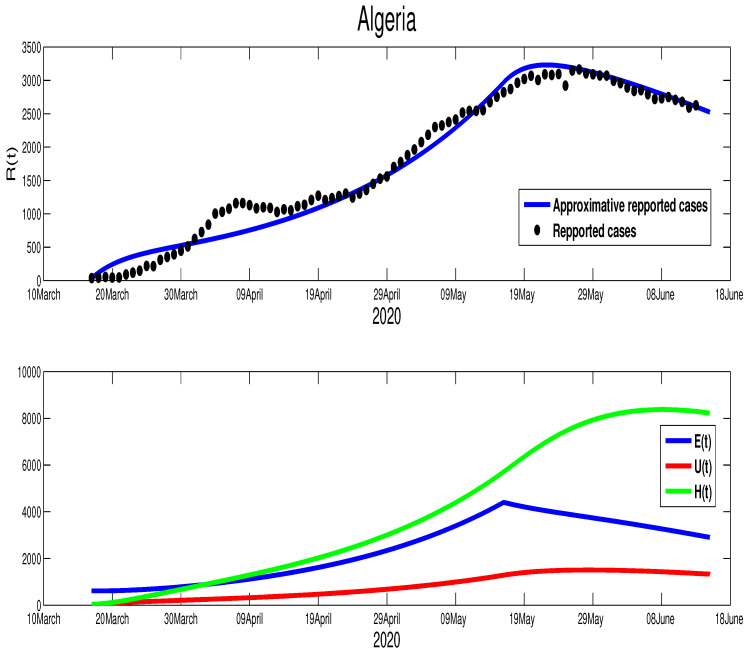
The data fitting of the reported COVID-19 cases in Algeria and prediction of the other variables of our model.

**Figure 5 biology-09-00373-f005:**
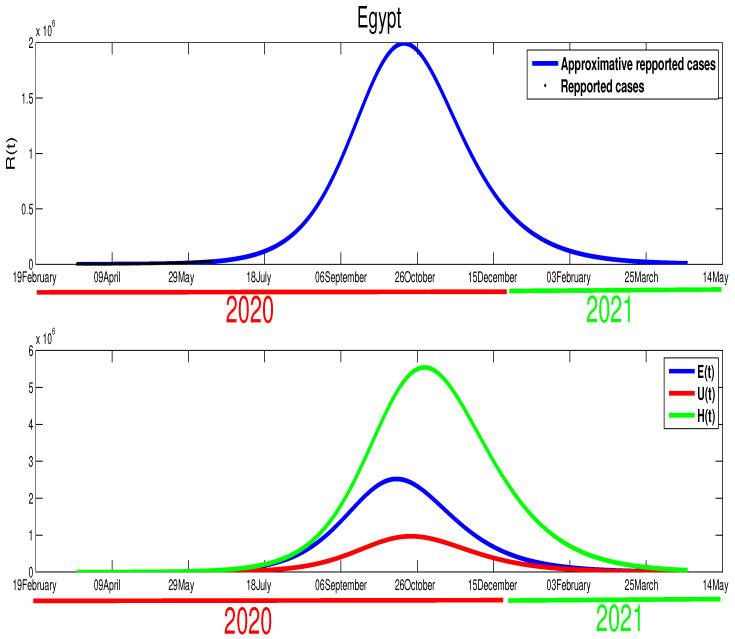
The predictive results for Egypt.

**Figure 6 biology-09-00373-f006:**
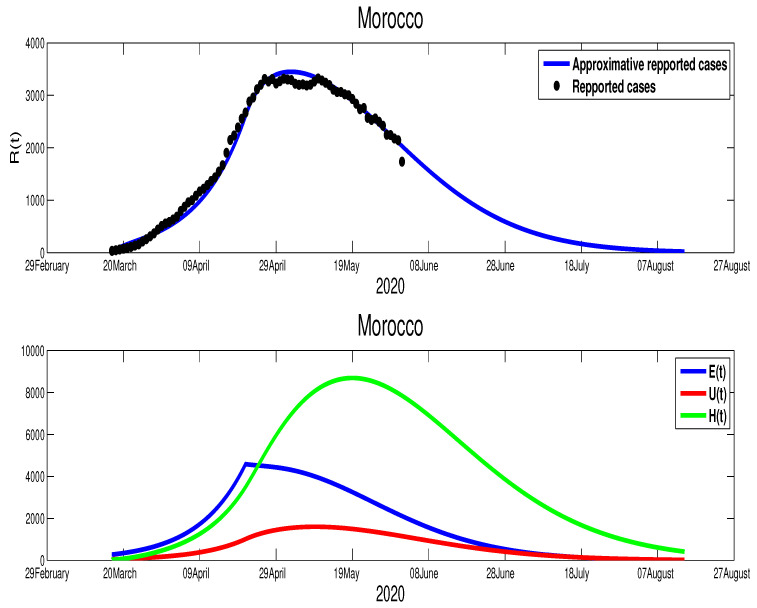
The predictive results for Morocco.

**Figure 7 biology-09-00373-f007:**
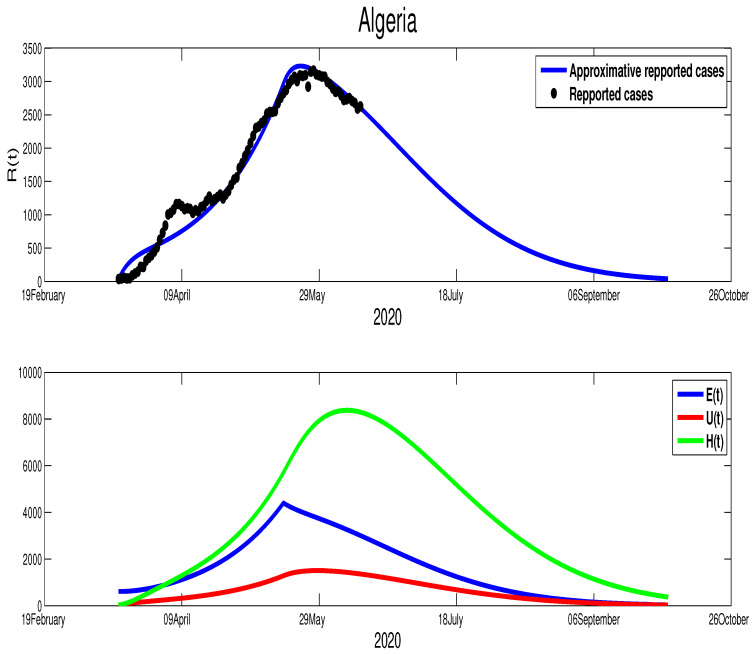
The predictive results for Algeria.

**Figure 8 biology-09-00373-f008:**
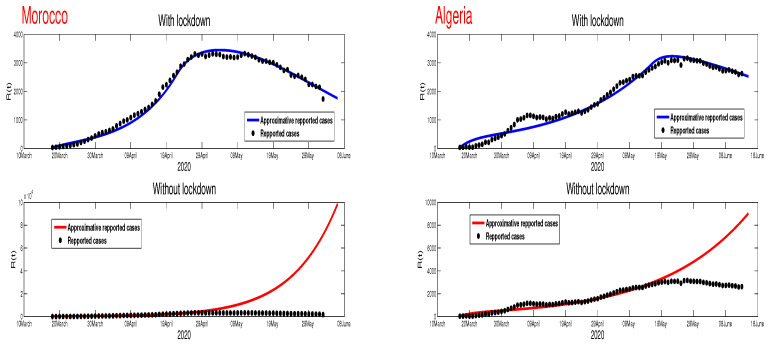
The time series of the number of reported infected cases with and without the lockdown, where it shows the importance of the lockdown in reducing the burden of the epidemic in Morocco and Algeria.

**Figure 9 biology-09-00373-f009:**
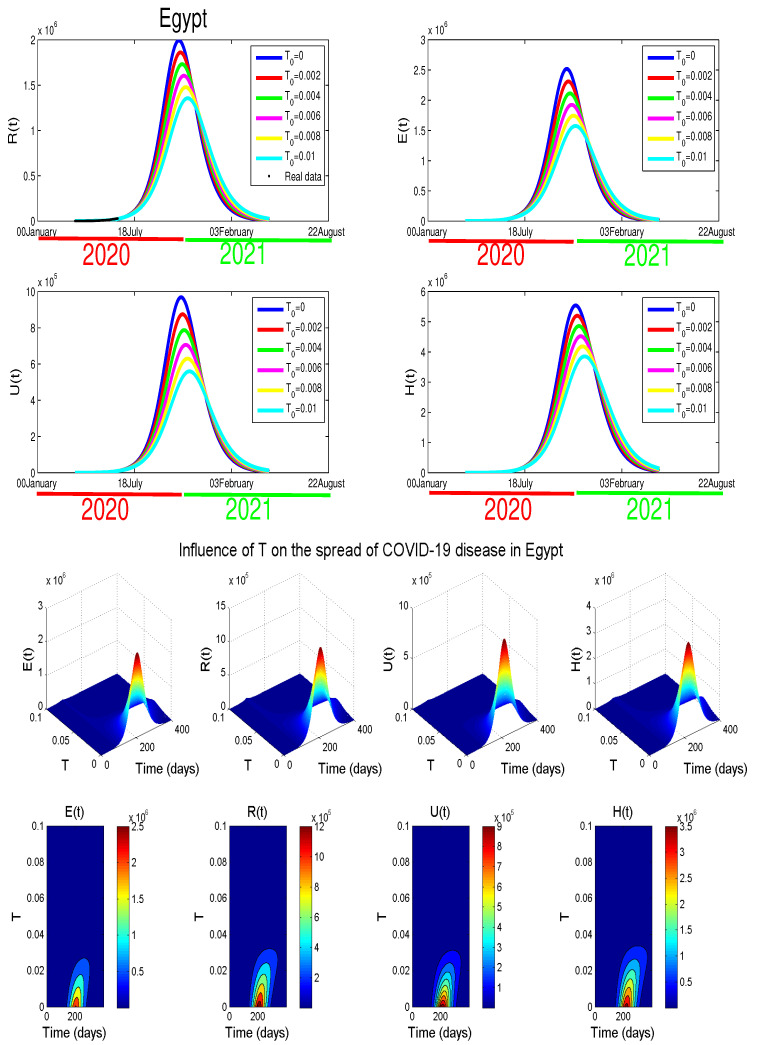
The impact of chaotic tests on the spread of the COVID-19 disease in Egypt.

**Figure 10 biology-09-00373-f010:**
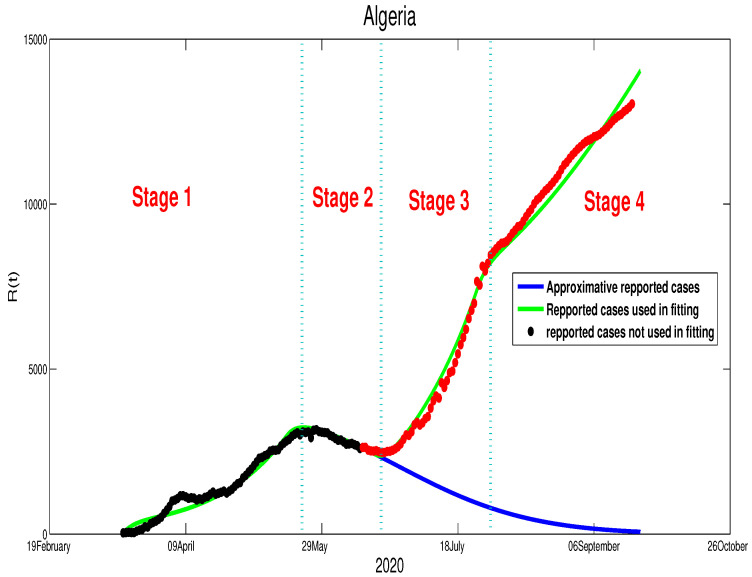
The new fitting of the data for the COVID-19 in Algeria after the relaxation of the population mobility restrictions.

**Figure 11 biology-09-00373-f011:**
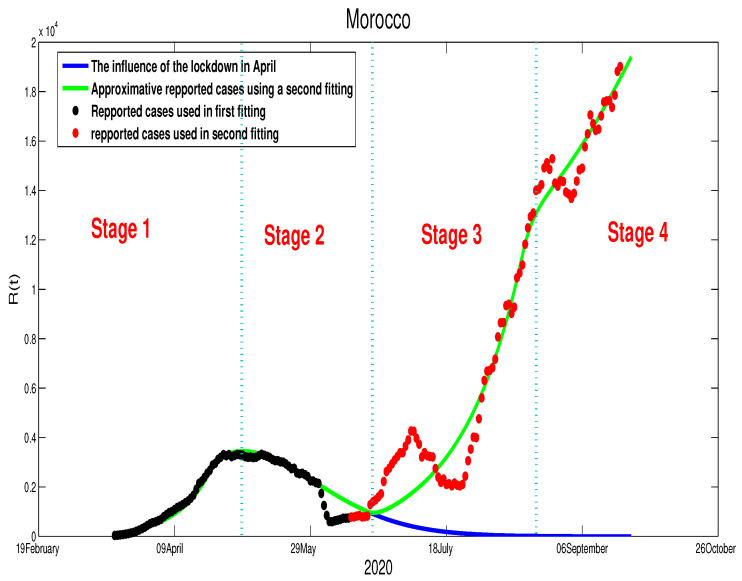
The new fitting of the data for COVID-19 in Morocco after the relaxation of the population mobility restrictions.

**Figure 12 biology-09-00373-f012:**
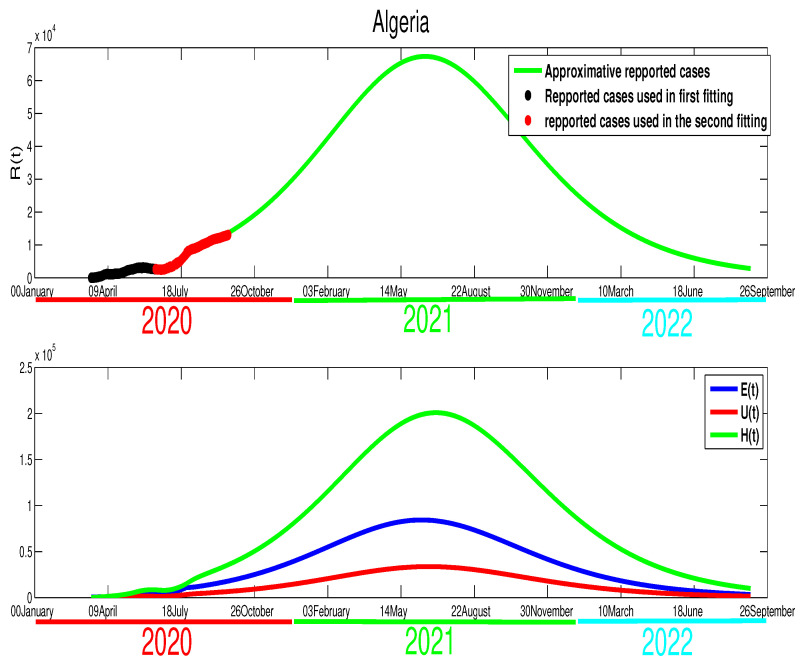
The projection of the number of COVID-19 cases in Algeria.

**Figure 13 biology-09-00373-f013:**
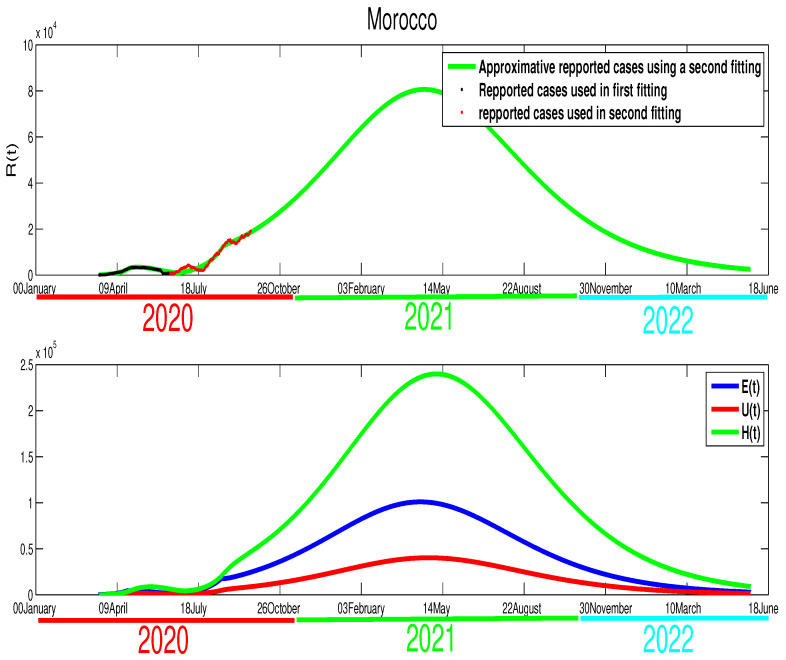
The projection of the number of COVID-19 cases in Morocco.

**Figure 14 biology-09-00373-f014:**
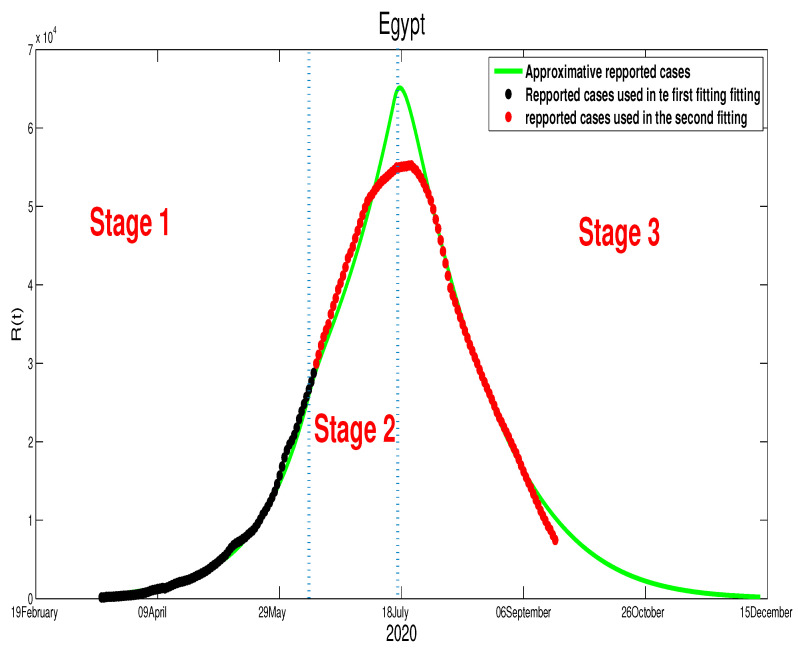
The evolution of the infection cases in Egypt after the relaxation of the population mobility restrictions.

**Figure 15 biology-09-00373-f015:**
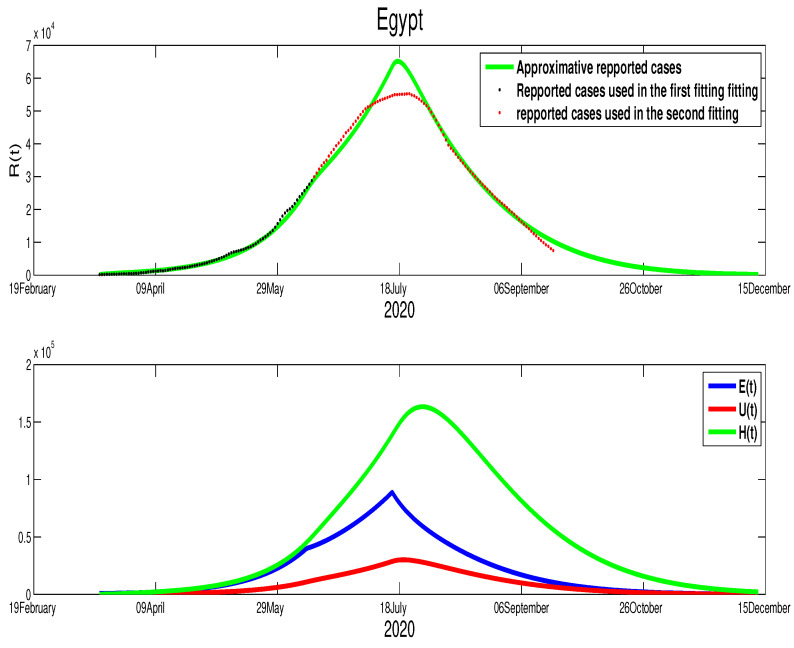
The evolution of the infection cases in Egypt until the end of 2020.

**Table 1 biology-09-00373-t001:** The basic reproduction number in each country during the different stages of the pandemic.

Country	Stage 1	Stage 2	Stage 3	Stage 4
**Egypt (** $R_{0}$ **)**	1.3641	1.1919		
**Morocco (** $R_{0}$ **)**	1.6141		1.3493	1.0971
**Algeria (** $R_{0}$ **)**	1.2859		1.2563	1.0819

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
