# Peer review of "Modeling the Impact of Unreported Cases of the COVID-19 in the North African Countries"

_biology, 2020, doi:10.3390/biology9110373_

Round 1

Reviewer 1 Report

The authors propose a mathematical model to investigate the impact of unreported cases of the COVID-19 in three North African countries: Algeria, Egypt, and Morocco. They use the mobility reports by Google and Apple to quantify the effect of the population movement restrictions on the evolution of the active infection cases. They also approximate the number of the population infected unreported and estimate the end of the epidemic wave. The authors also propose their model to estimate the second wave and to project its end.

Minor revisions:

  • some figures titles and legends are not readable. The authors need to improve the quality of the figures;
  • a few typos both in the text and in the figures need to be corrected.

Author Response

The authors propose a mathematical model to investigate the impact of unreported cases of the COVID-19 in three North African countries: Algeria, Egypt, and Morocco. They use the mobility reports by Google and Apple to quantify the effect of the population movement restrictions on the evolution of the active infection cases. They also approximate the number of the population infected unreported and estimate the end of the epidemic wave. The authors also propose their model to estimate the second wave and to project its end.

Minor revisions:

  • some figures titles and legends are not readable. The authors need to improve the quality of the figures;

Done! All the figures are reconstructed for better quality and with better legends.

  • a few typos both in the text and in the figures need to be corrected.

Done! We proofread the paper and made all the necessary change in blue color 

Reviewer 2 Report

The authors have given sufficient technical information towards the raised questions in their rebuttal and in the new revised version of the paper, and therefore I may now recommend acceptance. However, the language still needs improvement to be more easily readable and the figures are not yet of good quality for a journal publication.

Author Response

The authors have given sufficient technical information towards the raised questions in their rebuttal and in the new revised version of the paper, and therefore I may now recommend acceptance. However, the language still needs improvement to be more easily readable and the figures are not yet of good quality for a journal publication.

We thank the reviewer for her/his comment and for accepting our paper.

The paper wad proofread and we made all the necessary corrections as it can be seen in blue color.

For the figure, we reconstruct almost all the figures with better quality.

This manuscript is a resubmission of an earlier submission. The following is a list of the peer review reports and author responses from that submission.

Round 1

Reviewer 1 Report

This paper implements a SIRU-type model to investigate the impact of unreported cases of the COVID-19 in three North African countries: Algeria, Egypt and Morocco. In its present form this work should not published in MDPI Biology. The following comments are provided for improvement of the contribution for future resubmission:

  1. The authors do not provide any information on the mathematical estimation technique employed to reach the parameters values for each country.
  2. Also, the authors should have saved part of the available data for validating the model predictive features, after estimating parameters with a reduced number of data points.
  3. The authors mention employing the circulation data (Google and Apple) in the analysis. However, these data were only employed in a qualitative evaluation of the reported cases evolution in the three countries. It would be more meaningful if this information was applied directly in parametrizing the transmission rate, thus associating the circulation increase or reduction to the transmissivity.
  4. The authors have made two statements that are apparently conflicting with each other. As can be extracted from the two sentences below, circulation reduction was not markedly different between Egypt and Morocco, nevertheless the initial evolution of the disease was quite different in each country. It looks like circulation data alone is not enough to explain the differences in evolution, and the authors should provide a more deeper analysis on the remaining parameters or public health measures that could have influenced the behaviors (for instance, enforcement of masks use, closure or not of schools and children activities, which are not usually perceived by the circulation cell phone engines, etc).
  • Egypt: “Although the mobility report in [16] shows reduction of mobility -50% of mobility in public places and -2% in grocery and pharmacy . -26% in parks, -44% in transit stations and -13% in workplaces. However, there is +12% increase in the places of residence, which shows the effort of the Egypt to reduce the infection did not help in slowing down the increase of the number of cases.”
  • Morocco: “…we clearly see the mobility restriction implemented by this country had lead more reduction of population mobility compared to Egypt, with -54% in shops and leisure, -23% in food and pharmacies, -41% in parks, -51% in transit stations, -31% in workplaces and +20% for the places of residence.”
  1. As the authors have demonstrated, the predictive capability of the model is very limited, if not dynamically associated with the public health measures alterations along the epidemic evolution. Thus, as for the transmission rate, the authors should implement a time variable behavior in the testing capacity, instead of a single constant value at a certain prescribed time. For instance, this has been suggested by P. Magal and collaborators in the article: Cotta, R.M., C.P. Naveira-Cotta, and P. Magal, “Mathematical parameters of the COVID-19 epidemic in Brazil and evaluation of the impact of different public health measures”, MDPI Biology, Special Issue "Theories and Models on COVID-19 Epidemics", V.9, no.8, pp.220-246, 2020. doi: 10.3390/biology9080220
  2. As the authors have observed in the conclusions, the actual behavior of the COVID-19 evolution in all the three countries was markedly different from what the present model was able to predict employing the data up to mid June. This is the major point about not accepting the paper in its present form. The authors should now be able to modify the model accordingly so as to demonstrate that it could have been useful once the authors had implemented a scenarios analysis, accounting for possible future changes in societal behaviors and/or enforced public health measures. Without such an analysis, the final conclusions of the paper would be essentially against the use of such class of models in studying epidemics evolution, which would not be in fact a correct final appreciation. Therefore, a new version of these simulations should include different scenarios of evolution, by altering the transmission rate and testing capabilities, so as to encapsulate the actual behavior now made available.
  3. A substantial language revision is required. In its present form, I have found some difficulties in fully understanding part of the authors comments.
  4. The quality of the figures should be improved. The labels are difficult to read.

Author Response

We would like to thank the reviewer for her/his constructive report. As the reviewer can see, the reviewed version of our paper was improved substantially and add a new section that deals with the second wave of the pandemic in these countries.

The answers to the reviewer's comments are below.

  • The authors do not provide any information on the mathematical estimation technique employed to reach the parameters values for each country.

We fit the data of each country, from WHO to our modeling, using the well-known approach of the least square method. The method was mainly used to estimate the infection rate beta. The other parameters are estimated base on the available studies on modeling the COVID-19, as described in the introduction section.

  • Also, the authors should have saved part of the available data for validating the model predictive features, after estimating parameters with a reduced number of data points.

We do not think that this an issue because we had done this technique in our recent paper ( See Bentout et al. 2020). In fact, in this new version we went a step ahead even to fit the date after June 24th

  • The authors mention employing the circulation data (Google and Apple) in the analysis. However, these data were only employed in a qualitative evaluation of the reported cases evolution in the three countries. It would be more meaningful if this information was applied directly in parametrizing the transmission rate, thus associating the circulation increase or reduction to the transmissivity.

We used the mobility data to understand the nature of each country's adherence to the lockdown. This is an essential point in our comprehension of the spread of the disease from one country to another. We did not use the mobility date to estimate the parameter of the model. However, the understanding of mobility had helped us to choose the right beta for each country.

  • The authors have made two statements that are apparently conflicting with each other. As can be extracted from the two sentences below, circulation reduction was not markedly different between Egypt and Morocco. Nevertheless the initial evolution of the disease was quite different in each country. It looks like circulation data alone is not enough to explain the differences in evolution, and the authors should provide a more deeper analysis on the remaining parameters or public health measures that could have influenced the behaviors (for instance, enforcement of masks use, closure or not of schools and children activities, which are not usually perceived by the circulation cell phone engines, etc).
  • Egypt: “Although the mobility report in [16] shows reduction of mobility -50% of mobility in public places and -2% in grocery and pharmacy . -26% in parks, -44% in transit stations and -13% in workplaces. However, there is +12% increase in the places of residence, which shows the effort of the Egypt to reduce the infection did not help in slowing down the increase of the number of cases.”
  • Morocco: “…we clearly see the mobility restriction implemented by this country had lead more reduction of population mobility compared to Egypt, with -54% in shops and leisure, -23% in food and pharmacies, -41% in parks, -51% in transit stations, -31% in workplaces and +20% for the places of residence.”

We thank the referee for point out this very important point. In fact, the mobility data is the only way to measure how the population is adhering to the NPI measures, such as lockdown, that each country is implementing. Other NPI, such as the enforcement of masks use, cannot be measured. As far as “closure or not of schools and children activities”, it can also be measure by the mobility data since the children are assumed to be monitored by the adults. On the other hand, the increase of mobility in the residence shows how much the population stays within their residence and not traveling from place to place. The +12 % increase in the residential places in Egypt compared to Morocco, +20 % showed that the population in Morocco is more adherent to lockdown compared to Egypt.  Which means the population in Morocco have more restriction on mobility compared to Egypt.  We added a sentence to explain this point.

  • As the authors have demonstrated, the predictive capability of the model is very limited, if not dynamically associated with the public health measures alterations along the epidemic evolution. Thus, as for the transmission rate, the authors should implement a time variable behavior in the testing capacity, instead of a single constant value at a certain prescribed time. For instance, this has been suggested by P. Magal and collaborators in the article: Cotta, R.M., C.P. Naveira-Cotta, and P. Magal, “Mathematical parameters of the COVID-19 epidemic in Brazil and evaluation of the impact of different public health measures”, MDPI Biology, Special Issue "Theories and Models on COVID-19 Epidemics", V.9, no.8, pp.220-246, 2020. doi: 10.3390/biology9080220

We thank the reviewer for this valuable comment.  We are aware of the possibility of having the testing capacity as a time-dependent variable. However, we opt not to use it in work. Instead, we extended our paper, in the new version, to even fit the data after June 21th, which gave us the stage of the progress of the disease per each country. The new section shows that the relaxation of human mobility restriction is a measure that the country had to do to adapt with the progress of the disease.  We also add the reference that the reviewer suggested as it alien in the same direction of our work.

  • As the authors have observed in the conclusions, the actual behavior of the COVID-19 evolution in all the three countries was markedly different from what the present model was able to predict employing the data up to mid June. This is the major point about not accepting the paper in its present form. The authors should now be able to modify the model accordingly so as to demonstrate that it could have been useful once the authors had implemented a scenarios analysis, accounting for possible future changes in societal behaviors and/or enforced public health measures. Without such an analysis, the final conclusions of the paper would be essentially against the use of such class of models in studying epidemics evolution, which would not be in fact a correct final appreciation. Therefore, a new version of these simulations should include different scenarios of evolution, by altering the transmission rate and testing capabilities, so as to encapsulate the actual behavior now made available.

The intension of the authors was mainly to estimate the unreported case in the period up to June 24th, 2020. Our main goal was not to project the progress of the disease after that period. All the mathematical models are good in making short time estimation, and they all fail badly to give the long-term prediction. In the case of the North African Countries, the game-changer in the progress of the case after June 24th was the celebration of the Eid, which was on July 30th  and 31st , which one month before the Eid. In this period, the countries relaxed their measure, and people had losing up their guards, and the mobility had an increase, which affected the progress of the case in all the three countries.

As a response to the reviewer concerned about the feasibility of the model, we extend our work using the same model to fit the data of each country up to the date.

  • A substantial language revision is required. In its present form, I have found some difficulties in fully understanding part of the authors comments.

We made sure to proofread the paper by a native speaker to ease the reading of the article.

  • The quality of the figures should be improved. The labels are difficult to read.

DONE

Reviewer 2 Report

See attached file.

Author Response

We thank the reviewer for constructive comments. As the reviewer can see in our new version, we made a substantial improvement in our paper to answer all the comments. The answers the reviewer report are below 

  1. It is true that the reduction rate has an important role in model construction, and it is wise to consider it in the mathematical approximation. But, the incubation period varies between 3 days and 14 days and there is no sufficient information on value of this rate. It is confirmed by the WHO that COVID-19 disease can be transmitted in this stage through a droplet left in surfaces by the person in this stage, but the percentage or the probability (value of ) still unknown. Hence, we considered that the infected person and person in the incubation stage have the same degree of transmission (=1). Moreover, we refer the reviewer to the work of Z. Liu , P. Magal . O.Seydi . G. Webb , biology, 2020. Where the authors used a similar argument to model the unreported cases.
  2. We thank the reviewer for catching this typo mistake. Fix it!
  3. We thank again the reviewer for point out this error. In fact, our Ro was correct and we made a typo mistake ( type copy-paste) in the expression of V and F.
  4. Add a table to answer this question, we give the variance in 95%CI
  5. The authors use such terminology because of a lack of mobility data on Algeria in Google and Apple data. In fact, Algeria and Morocco have almost the same population and the two populations are similar in many aspects of their lifestyle as well as their attitude toward the lockdown. We find that this an optimistic assumption as we do not have any alternative.
  6. The typos mistakes and language of the paper was reviewed by a native speaker to fix all the comments of the reviewer.

Reviewer 3 Report

In this work, the authors study a mathematical model that investigates the impact of unreported cases of the COVID-19 in three North African countries and produce mobility reports that help to quantify the effect of the population movement restrictions on the evolution of the active infection cases.

The topic is relevant and interesting because one of the major issues during this pandemic is to trace people who got in touch with COVID-19 infected people, in order to stop the spread of the virus.

The topic is not completely original indeed the authors themselves cite that there are others articles predicting the reported and unreported cases of COVID-Q9 in other countries. Despite that the originality of the work lays in taking in consideration the mobility of the population using mobility data provided by the COVID-19 Community Mobility Reports, Google- 2020 and the Mobility Trend Reports, Apple- 2020. These data allow investigating the effect of the measures taken by each country, to restrict the population movement, on the reduction of the impact of the unreported cases.

The manuscript is well written and the story-line is clearly exposed and easy to read. Please just check some typos present in the text.

The authors address the main question they posed. Indeed they conclude that by looking to the mobility data of the considered countries they were able to analyse the efficacy of the lockdown measure by using the mathematical model they developed, and the conclusions are consistent with the data presented in the manuscript.

In this work, the authors examined a mathematical model that investigates the impact of unreported cases of the COVID-19 in three North African countries and produce mobility reports help to quantify the effect of the population movement restrictions on the evolution of the active infection cases

The manuscript is well written and the story-line is clearly exposed.

The State of the Art section is too short. Indeed, it is less than one page in length and is inserted in a more general paragraph concerning the model construction. Moreover, there is a lack of contents especially about the implementation of mathematical models. In the light of these considerations, I strongly suggest to expand this part and cite all the papers that can be found in the related fields, in order to give a robust background to the work. In general, the references section should be severely improved, especially when a manuscript explores for the first time new field.

As an example, I suggest you to cite the following:

  • Improta, G., Russo, M.A., Triassia, M., Converso, G.,Murino, T., Santillo, L. C. Use of the AHP methodology in system dynamics: Modelling and simulation for health technology assessments to determine the correct prosthesis choice for hernia diseases. Mathematical Biosciences Volume 299, May 2018, Pages 19-27. doi.org/10.1016/j.mbs.2018.03.004.

The experimental section is robust and well written and the conclusions are consistent with the produced data.

Author Response

We thank the reviewer for her/his nice report. 

We followed all the comments made and added the suggested reference. 

Reviewer 4 Report

In the manuscript “Modeling the impact of unreported cases of the COVID-19 in the North African Countries”, the authors study a mathematical model that investigates the impact of unreported cases of the COVID-19 in three North African countries: Algeria, Egypt and Morocco. They use the mobility reports from Google and Apple to understand how the population respect the restriction implemented in each country. They suggest that these mobility reports help in approximating the number of people infected unreported, those that need hospitalization and estimate the end of epidemic wave. Finally, they also suggest some additional measures that can be considered to reduce the burden of SARS-CoV-2.

The manuscript needs to be revised to correct some grammatical errors and a few typos.

The references are appropriate.

Major revisions:

  • The authors use sometimes SARS-CoV-2 and COVID-19 as synonymous. They are not: SARS-CoV-2 is the novel coronavirus, while COVID-19 includes all the medical conditions caused by SARS-CoV-2. I would suggest the authors to review the manuscript and use the correct terminology.
  • The authors should add more information related to their model in a dedicated section. Which mathematical model they use? Is it different from other models used before, or simply do they take into consideration the mobility data?
  • The authors should improve the quality of the figures: titles and legends are not very legible. They should also include dates (not just the Time) to make them easier to follow and understand.
  • Are the data up to date? The authors consider the active cases of infection until June 14th. The authors should add more data to increase the accuracy of their model.
  • The authors should revise the situation in each country, in particular in Morocco. In page 7 and in section 4 the authors declare that “the epidemic in Morocco would have end by August 7th 2020” and “the infection will disappear from the Moroccan community by August 7th, 2020”. As of today the virus is still actively spreading and there are 17632 active cases (https://www.worldometers.info/coronavirus/country/morocco/). See also fig. 10 for a revised discussion.
  • How do the authors fit in their model other parameters, such as temperature, population density, demographic….?
  • a few typos and grammatical errors need to be corrected.

Author Response

We would like to thank the reviewer for her/his constructive comments that help to improve the quality of our work. As the reviewer can see in the new version, we extended our work to the second wave of the pandemic. 

The answers to the reviewer's comments are below.

  • The authors use sometimes SARS-CoV-2 and COVID-19 as synonymous. They are not: SARS-CoV-2 is the novel coronavirus, while COVID-19 includes all the medical conditions caused by SARS-CoV-2. I would suggest the authors to review the manuscript and use the correct terminology.

We did not use even in a single occasion  “SARS-CoV-2” in our paper. We used only “COVID-19”. We do not know what the reviewer is referring to.

  • The authors should add more information related to their model in a dedicated section. Which mathematical model they use? Is it different from other models used before, or simply do they take into consideration the mobility data?

We thank the reviewer for his comment. We already add enough information on the nature of our model compared to other models. We also explain that the use of the mobility data was limited in helping us to choose the beta ( the rate infection) and to fit the data of each country

  • The authors should improve the quality of the figures: titles and legends are not very legible. They should also include dates (not just the Time) to make them easier to follow and understand.

Done!

  • Are the data up to date? The authors consider the active cases of infection until June 14The authors should add more data to increase the accuracy of their model.

We did add more data up to September 29th, 2020, and fitted our model to this new data.

  • The authors should revise the situation in each country, in particular in Morocco. In page 7 and in section 4 the authors declare that “the epidemic in Morocco would have end by August 7th 2020” and “the infection will disappear from the Moroccan community by August 7th, 2020”. As of today the virus is still actively spreading and there are 17632 active cases (https://www.worldometers.info/coronavirus/country/morocco/). See also fig. 10 for a revised discussion.

The new section of our paper deals with this issue.

  • How do the authors fit in their model other parameters, such as temperature, population density, demographic….?

All our initial condition was based on the demography of each country. Dealing to population density required different modeling approaches which not in our current scope.

  • a few typos and grammatical errors need to be corrected.

Done!